# Photopolymer Holographic Lenses for Solar Energy Applications: A Review

**DOI:** 10.3390/polym16060732

**Published:** 2024-03-07

**Authors:** Eder Alfaro, Tomás Lloret, Juan M. Vilardy, Marlón Bastidas, Marta Morales-Vidal, Inmaculada Pascual

**Affiliations:** 1Grupo de Investigación en Física del Estado Sólido (GIFES), Faculties of Basic and Applied Sciences, and Engineering, Universidad de La Guajira, Riohacha 440007, La Guajira, Colombia; ealfaro@uniguajira.edu.co (E.A.); jmvilardy@uniguajira.edu.co (J.M.V.); 2Departamento de Óptica, Farmacología y Anatomía, Universidad de Alicante, Carretera San Vicente del Raspeig s/n, 03690 San Vicente del Raspeig, Spain; marta.morales@ua.es (M.M.-V.); pascual@ua.es (I.P.); 3Grupo de Investigación Desarrollo de Estudios y Tecnologías Ambientales del Carbono (DESTACAR), Faculty of Engineering, Universidad de La Guajira, Riohacha 440007, La Guajira, Colombia; marlonjoseb@uniguajira.edu.co

**Keywords:** photopolymers, holographic lenses, solar energy, electrical energy, thermal energy

## Abstract

Holographic lenses (HLs) are part of holographic optical elements (HOE), and are being applied to concentrate solar energy on a focal point or focal line. In this way, the concentrated energy can be converted into electrical or thermal energy by means of a photovoltaic cell or a thermal absorber tube. HLs are able to passively track the apparent motion of the sun with a high acceptance angle, allowing tracking motors to be replaced, thus reducing the cost of support structures. This article focuses on a review of the materials used in the recording of a holographic lens (HL) or multiple HLs in photovoltaic and/or concentrating solar collectors. This review shows that the use of photopolymers for the recording of HLs enables high-performance efficiency in physical systems designed for energy transformation, and presents some important elements to be taken into account for future designs, especially those related to the characteristics of the HL recording materials. Finally, the article outlines future recommendations, emphasizing potential research opportunities and challenges for researchers entering the field of HL-based concentrating solar photovoltaic and/or concentrating solar thermal collectors.

## 1. Introduction

In most ambitious climate scenarios and innovative improvements in energy efficiency make solar energy one of the most appropriate energy resources to be inexhaustible and readily available on our planet. Only a small fraction of the solar radiation on Earth can meet the world’s energy demand in the future, the available technologies in sunlight conversion are as follows: photovoltaic (PV), photothermal, and photochemical; recent research shows many advances in these technologies and especially photovoltaic that currently this has become the third most important renewable energy source after hydro and wind [1,2,3].

On the other hand, photothermal energy is also growing significantly, four main technologies have been developed that use solar irradiation concentrating on a thermal working fluid, these technologies concentrate solar irradiation through the principles of reflection or transmission of optical materials such as parabolic trough systems, solar tower systems, solar dish Stirling systems and linear Fresnel systems [4]. These can be divided according to the application needs to be related to the temperatures reached by the working fluids, such as non-tracking light concentration, single-axis tracking, and dual-axis tracking; the latter is often used in systems with high-temperature requirements, such as parabolic dishes used in power towers or heliostat arrays [5]. These technologies allow high temperatures to be reached in a working fluid, where the heat generation process starts with the absorption of thermal energy in the receiver, achieving very high temperatures, ranging from 100 °C to several thousand degrees Celsius, this thermal energy is used to generate steam and run a steam Rankine cycle, with the aim of producing electricity [6,7,8].

Since the 1970s, hybrid photovoltaic/thermal (PVT) technologies have been proposed, which have combined heat and power (CHP) energies. Hybrid flat-plate PVT systems do not require concentrated solar radiation and are designed to extract electrical and thermal energy from PV cells. Typically, water or air is used as the coolant, and heat transfer fluid (HTF) flows through the plates. The total cogeneration efficiency of these flat plate PVT systems is consistently higher than that achieved using two separate systems. In addition, the integration of PV and thermal collectors can reduce production and installation costs, making it more economical and practical for applications requiring both electricity and heat (such as building-integrated installations) [9,10].

Sporadically designed physical systems using HOE have been implemented to increase the energy efficiency of solar energy transformation methods by concentrating energy at a specific point or line. An optical efficiency of 93% and an adjustable subsystem output power ratio is achieved by using HOE in a PVT system. This allows for energy storage and the ability to manage solar intermittency [11]. Most HLs employed include optical components such as lenses, mirrors, and gratings, all of which are volumetric phase holograms. For a specific wavelength and direction, volume phase holograms can achieve close to 100% efficiency. Three key attributes of holograms are angular selectivity, diffraction, and scattering. These characteristics are highly dependent on the holographic recording materials and the recording geometry [12].

This article reviews recent research with the purpose of better understanding the parameters or variables to optimize the performance of holographic optical elements for holographic solar concentrator applications, in the following order: In Section 2, basic concepts of HLs are presented. In Section 3, the materials used for the construction of HLs are presented. In Section 4, the photopolymers used in HLs for solar spectral concentration applications are shown. In Section 5, a discussion is made, and finally, in Section 6, conclusions are given.

## 2. Basic Concepts of HLs

Holography is based on Gabor’s principle of reconstructing the interference of two wavefronts [13]. It is a technology that allows the recording and reconstruction of any object. Within the holography, we find the holographic optical elements (HOE) that by their applications make it interesting, because they replace curved and bulky refractive optical elements with simple, flat, and light elements [14]. The HOE stores the interference pattern produced by two spatially superimposed coherent beams, this pattern creates a photonic structure that diffracts the light in the desired way.

The first HOE concept of a holographic application as a holographic mirror was described by Denisyuk in 1962 [15]. And, the construction of the first point focusing on HL was demonstrated by Schwar et al. in 1967 [16]. Given that breakthrough at the time, other researchers proposed to gather several functions on a single substrate by multiplexing two or more HOE [17,18,19,20,21,22,23] due to their high diffraction efficiency and narrow band frequency characteristics.

### 2.1. Necessary Conditions of HLs in Solar Energy Harvesting Application

Solar applications require efficient collection of photons propagating from a source of apparent motion (the sun). The sun can be considered as an approximate blackbody (T∼6000) K. As such it emits, randomly polarized light over a broad spectral bandwidth, with the highest power emitted between ∼350 and 2000 nm and a maximum emission wavelength of around 480 nm. The apparent motion of the sun is from east to west during the day and follows a higher and lower path at different times of the year, see Figure 1. In addition, the atmosphere affects the spectrum and nature (direct or diffuse) of the light reaching the solar concentrators on the Earth’s surface. These properties place considerable demands on optical components and systems to achieve high optical efficiency.

Given the nature of sunlight illumination, HLs must be able to randomly diffract polarized light over a wide spectral band and different angles of incidence. It is difficult to meet all of these requirements simultaneously; however, it is possible to design HLs that improve the performance of concentrating systems in a cost-effective manner. For these designs, the parameters taken into account in the designs are: angular selectivity, dependence of wavelength selectivity on fringe spatial frequency, and diffraction efficiency [24].

### 2.2. Setups for HLs Recording and Reconstruction

For HL development, diffraction efficiency remains the limiting factor in the acceptance angle range. To improve these aspects, theoretical and experimental models have been proposed to obtain higher angular selectivity and thus high diffraction efficiency. The goal is to capture sufficient energy from the solar spectrum without tracking the apparent motion of the sun. Within these mounts, there is a register of one or more spherical lenses and a register of cylindrical lenses, in some cases the register mounting is on the same axis or off-axis (symmetrical and asymmetrical) [25,26,27,28].

Depending on the recording presented in the hologram, HLs can be of two types: spherical or cylindrical. For spherical HLs, one of the wavefronts of the recording beam corresponds to a spherical wave in the interference pattern, while the reference beam corresponds to a plane wavefront, thus concentrating the incident light in the reconstruction at an output point. For cylindrical HLs, if the interference pattern of one of the recording beam wavefronts corresponds to a cylindrical wave and the reference beam corresponds to a plane wavefront, then the diffracted light will be concentrated into a line at the output of the hologram.

### 2.3. Recording Stage

In the recording stage, HLs are obtained from the interference of a plane beam (reference beam) and a convergent (positive HLs) or divergent (negative HLs) beam depending on the type of HL. On-axis and off-axis recordings can be considered, and in the latter, symmetrical and asymmetrical recordings can be differentiated.

### 2.4. Reconstruction Stage

On the one hand, in the reconstruction stage, positive HLs are reconstructed with the reconstruction beam incident on the hologram from the side where the recording material is located. On the other hand, the negative HLs are reconstructed with the conjugated beam incident on the hologram from the opposite side of the recording material. The angle of the reconstruction beam for the different lasers has been calculated using Bragg’s law
(1)$$\sin\left(\theta \right)=\frac{\lambda }{2\Lambda }$$
where 2θ is the angle between beams ($2\theta =\theta_{O}-\theta_{R}$), $\theta_{O}$ is the angle of the object beam with respect to the normal to the plate and $\theta_{R}$ is the angle of the reference beam with respect to the same normal, taking the positive angles clockwise and the negative angles counterclockwise; λ is the wavelength, and Λ is the period of the interference pattern. In addition, the expressions for the calculation of the image angle and focal length of the HLs are given by the following equations [29,30].
(2)$$\sin\left(\theta_{I}\right)=\sin\left(\theta_{C}\right)+\mu \left(\sin\left(\theta_{O}\right)-\sin\left(\theta_{R}\right)\right),$$
(3)$$\frac{1}{f_{HL}^{\prime }}=\frac{1}{R_{C}}+\mu \left(\frac{1}{R_{O}}-\frac{1}{R_{R}}\right),$$
where I,C,O,R are the subscripts of the image, reconstruction, object, and reference beams, respectively; and *R* is the distance from the source point to the hologram, respectively. Considering that the reference and reconstruction beams are plane waves, $R_{R}$ and $R_{C}$ tend to infinity, and $R_{O}$ is the refraction focal length. Furthermore, $f_{HL}^{\prime }$ is the focal length of each lens, while μ is the ratio of the reconstruction wavelength to the recording wavelength $\mu =\lambda_{C}/\lambda_{O}$.

The following Figure 2, Figure 3 and Figure 4 depict the optical schematic setup for the HLs recording and reconstruction using the on-axis and off-axis configurations.

## 3. Materials Used for the Construction of HLs

There exists a wide variety of hologram recording materials whose optical properties, such as their refractive index or absorption coefficient, can be modified as a function of the recording intensity of the two coherent light beams that produce an interference pattern in the recording material. When a hologram is recorded in a material as an absorption modulation, it is called an amplitude hologram because it is the amplitude term of the light wave that is affected when it interacts with the medium. When the refractive index and/or thickness of a material changes by recording, it is called a phase hologram. In the latter case, there is no absorption of incident light and the efficiency of the hologram can be higher than in the case of amplitude holograms [31,32,33,34].

To characterize a holographic recording material, it is necessary to understand its properties; these properties can determine the application in which it is intended to be used. Holographic recording materials can be divided into three categories. The first category is permanent recording materials, whose diffraction patterns cannot be erased or updated. The second category is renewable recording materials; you can erase the diffraction pattern and then reuse the material to record a new diffraction pattern. The third category is electronic components, which allow dynamic visualization of diffraction structures, because of their ease of use and continuous improvements in spatial bandwidth products, making them increasingly important in the research and development of dynamic diffractive optics [35].

The characteristics that define the performance of holographic recording material are spectral sensitivity, dynamic and linear response range, resolution or spatial frequency response, material change after processing, multiplexing capability, processing requirements, stability, maximum efficiency, and low toxicity [35,36,37].

**Spectral sensitivity** specifies the energy required at a given wavelength at which the material exhibits one or more responses. The response(s) of the material usually produce a change in the modulation of the refractive index, ∆n, for a phase grating or in the absorption, ∆α, of the material for an amplitude grating.

**Dynamic and linear response range** provides the maximum index or absorption modulation that can be achieved with the material. Most often, a linear response of the material modulation is desired, as this results in high-fidelity holograms.

**Resolution or spatial frequency response** is the number of grating period lengths that can be formed in the material. To design a hologram, it must be checked especially when the grating period is very small or large, to make sure it falls within the response range of the material, one way to calculate this is with the grating equation
(4)$$\Lambda =m\frac{\lambda_{0}/n}{\sin\left(\theta_{d}\right)-\sin\left(-\theta_{i}\right)},$$
where Λ is the fringe spacing, *m* is the diffraction order, $\lambda_{0}$ represents the recording wavelength, $\theta_{d}$ denotes the diffraction angle and $\theta_{i}$ is the angle of incidence, both angles being defined by the recording geometry.

**Changes in the material after processing:** When the material is processed after exposure, it may undergo changes in the physical dimensions of the holographic material, with post-exposure processing it may undergo changes in the average absorption and refractive index, resulting in aberrations and a decrease in the diffraction efficiency of the reconstructed holographic images.

**Multiplexing capability:** Multiplexing of holograms on the same recording material is often required for applications such as holographic data storage and passive concentrating solar trackers that act as converging lenses. Multiple gratings can be formed by recording several different holograms sequentially or by recording multiple beams of objects simultaneously in a single exposure. It is important to note that the multiplexing technique has been commonly used in data storage, but in recent years it has been applied to design holographic concentrators due to its great versatility. In order to use the multiplexing technique correctly, the recording material must be optimized, increasing the thickness in order to increase the dynamic range [38,39,40].

**Processing requirements:** For some materials, after recording they are processed to achieve the desired effects in the resulting holograms, e.g., some of the newer photopolymers are processed with a simple white light fixation and are much more practical for different applications.

**Stability** allows the hologram to persist over long periods of time with repeated use, under environmental conditions such as temperature, humidity, and exposure to the full solar spectrum, e.g., in the application of a PVT system.

**Maximum efficiency (η)** is the maximum diffraction expressed as the ratio of the first-order diffraction intensity to the incident intensity (external efficiency) or as the ratio of the first-order diffraction intensity to the hologram-free transmission intensity (internal efficiency). It should be noted that the internal efficiency is always higher than the external efficiency because absorption, scattering, and Fresnel reflection of the material have reduced the transmitted intensity. The efficiency has no unit and is usually expressed as a percentage (%).

**Low toxicity** reagents that are considered to be environmentally friendly, which may be biodegradable or have less environmental impact.

### 3.1. Transmission and Reflection Hologram

Among holograms, a distinction is made between those that work by transmission and those that work by reflection. In a transmission hologram, the incident light passes through the medium and is diffracted to the other side of the material, whereas in a reflection hologram, the light is diffracted to the same side of the material as the incident beam. Transmission holograms are angularly selective but suffer from spectral dispersion. For example, when a transmission hologram is illuminated with white light, the incident light is diffracted into a rainbow (the spectrum is spread out), whereas if it is illuminated with a monochromatic light source, diffraction occurs at a single angle of incidence (the Bragg angle). Reflection holograms are the opposite: they are spectrally selective and angularly tolerant, and if illuminated with white light, they return only the diffraction of the color of the beam used in the recording, acting as a filt er (spectral selection). However, reflection holograms can diffract incoming light over a large angle of incidence [41]. The optical scheme for the reflection HLs recording and reconstruction are shown in Figure 5.

### 3.2. Criteria for a Thick or Thin Hologram

Holographic lenses can be classified according to the thickness of the recording medium, and the spatial frequency recorded in this medium, so we would have thin holograms, where the thickness of the hologram is small compared to the distance between fringes, (they work according to the Raman–Nath diffraction regime), and volume or thick holograms, where the distance between the fringes is much smaller than the thickness of the medium, (they work according to the Bragg diffraction regime).

The classical distinction between these two types was made through the *Q* parameter.

In this classification we have that:If Q<1 they are considered thin holograms.If Q>10, they are considered volume holograms.If 1<Q<10, it is said to work in intermediate regime.

### 3.3. Kogelnik’s Coupled Wave Analysis (KCWA)

The most widely used theory for fitting diffraction efficiency curves and optimizing conditions in the design of holographic optical elements is KCWA [42]. This theory provides an analytical solution and its expression for the diffraction efficiency fit is represented in the Equation (Equation 5). This theory predicts the diffraction efficiency for a transmission phase grating with a thickness (*d*) around the Bragg angle at a specific $\lambda_{C}$.
(5)$$\eta =\frac{\sin^{2}{\left(\nu^{2}+\xi^{2}\right)}^{1/2}}{1+\xi^{2}/\nu^{2}}$$
Equation (Equation 6) shows how the variable ξ varies with the detuning parameter (ϑ), dependent on Λ (grating period), $\theta_{c}$ (reconstruction angle), $\theta_{c}^{\prime }$ (reconstruction angle inside the material), $\lambda_{c}$ (reconstruction wavelength), φ (slope of the fringes), and *n* (refractive index). All of these factors are shown to be present in Equation (Equation 7). When the system is functioning outside of the Bragg condition, ξ takes on a value of either positive or negative; otherwise, it assumes a value of zero. Because the diameter is not usually large, the lenses are normally fitted taking into account the center value of *K* (the grating vector).
(6)$$\xi =\frac{\vartheta d}{2\left(\cos\left(\theta_{c}^{\prime }\right)-\frac{K}{\beta }\cos\left(\varphi \right)\right)}$$
(7)$$\vartheta =K\left[\cos\left(\varphi -\theta_{c}\right)-\frac{K}{4\pi n}\lambda_{c}\right]$$

KCWA theory predicts that two strategies can be followed to obtain a wide acceptance angle in volume HLs: (a) reduce the spatial frequency (see Figure 6a), (b) reduce the thickness of the photopolymer film (see Figure 6b). These strategies contribute to being closer to the intermediate regime between volume holograms and thin holograms.

### 3.4. Overview of the Most Commonly Used Recording Materials for HLs

Permanent recording materials include silver halide, dichromated gelatine, photopolymers, photothermo-refractive glasses, holographic sensors, photoresist, and embossed holograms. For renewable recording materials, there are photochromic materials, polarization-sensitive materials, photorefractive materials, photothermoplastic process materials, and persistent spectral hole burners. And finally, those using electronic components; such as the focal plane array detector, acousto-optic modulator, and the spatial light modulator [35]. Three of these materials stand out for their applications in solar energy transformation: silver halide, dichromated gelatine, and photopolymers. For this reason, only these materials are detailed in this review, showing an overview of the main advances, properties, and their limitations, and emphasizing mainly photopolymers for being the most novel and promising materials in the use of solar energy applications [43].

#### 3.4.1. Silver Halide

Silver halide was the first material used to record holograms; at the time it was one of the most important materials for holography due to its numerous scientific and artistic applications, mainly because of its high sensitivity, it can record amplitude and phase holograms, it has a high power resolution and it is easy to implement. However, it also has some disadvantages: it is absorbent, it requires wet processing, it can create printing problems in phase holograms, etc. [44,45,46]. Their chemical composition is given by AgBr, AgI, and AgCl, gelatine (protein extracted from animal skin or bone collagen), sensitizing dyes, and stabilizers, they are called silver halide emulsions and are part of the holographic recording material. This holographic material has a sensitivity of a few ($10^{-5}$ mJ/cm^2^) and can reach spatial frequencies of 6000 lp/mm.

Silver halide emulsions require chemical processing to display holograms. Figure 7 shows the chemical process that converts silver halide crystals into holograms. These holograms can be either amplitude-modulated or phase modulated. For holographic lens recording applications, phase modulation is suitable to increase the diffraction efficiency up to 100%.

Good developing and bleaching processes for making reflection holograms also tend to work well for transmission holograms. They allow multiplexing holographic lenses by equal exposures of sequential recording on a silver halide emulsion, and the diffraction efficiency of each successive hologram will decrease; this effect is called “failure of the holographic reciprocity law” and results from the time delay between recordings. The effect applies to any type of holographic recording that is to be multiplexed [47]. A useful parameter for evaluating the diffraction efficiency of recording material in multiplexing applications is the dynamic range of the holographic material, which provides a numerical value for comparing different materials and represents the dynamic response of the material to holographic exposure [48].

#### 3.4.2. Dichromated Gelatin, DCG

T.A. Shankoff first used DCG to record holograms in 1968 [49]. Since then, DCG has proven to be a versatile recording material and is used in various applications requiring holograms with high diffraction efficiency and optical quality. DCG can be used on rigid or flexible substrates and flat or curved surfaces. The properties obtained by the holograms can be controlled by adjusting the material properties, exposure conditions, and gelatine processing methods, which provides great design flexibility [50,51,52,53]. It is interesting to note that there are also researchers who worked with silver halide and by chemical processes obtained DCG [54,55,56].

Compared to silver halide emulsions, DCG has a lower energy sensitivity, but a higher efficiency and a considerably higher resolution. The high refractive index modulation capability, high diffraction efficiency, high resolution, low noise, and high optical quality make DCG an almost ideal recording material for volume phase holograms [57]. However, it is very sensitive to environmental changes and requires chemical post-processing. In addition, DCG contains ammonium dichromate (AD), which is reported to be carcinogenic, mutagenic, and toxic to reproduction [58,59,60].

#### 3.4.3. Photopolymers

In 1969 photopolymers were first used as a holographic recording material by Close et al. [61]. exhibiting good photosensitivity properties, wide dynamic range, good hologram stability, and relatively low cost. All these virtues have made photopolymers the most promising materials for holographic applications. The development of photopolymers used to etch holograms generally consists of one or more monomers, a lightweight analog system, and an inactive component called a binder. Sometimes other components are added to control various properties, such as the lifetime and viscosity of the recording medium [62]. Figure 8 shows the process of recording a hologram on a photopolymer.

Table 1 presents some important characteristics of the three recording materials (silver halide, DCG, and photopolymers) most frequently used in the production of HLs in solar energy concentration, such as maximum diffraction efficiency, acceptance angle, material stability, multiplexing capability, spectral sensitivity for recording, and spectral response over a range of wavelengths.

## 4. Photopolymers Used in HLs for Concentrating Solar Spectrum Applications

Photopolymers usually consist of monomers, photosensitizing dyes, and initiators, either in liquid layer systems or in dry layer systems, which allows diversity in this type of material. Photopolymers can be classified into photopolymers with hydrophobic binders, such as polyesters, or methacrylic acid, and photopolymers with hydrophilic binders, such as polyvinylalcohol (PVA) or gelatin, including hybrid materials made by the sol-gel process [74,75].

Photopolymers have proven to be one of the promising technologies for holographic records, due to the ease of processing the materials and low costs, in addition, their properties allow them to be implemented in several applications, such as data storage, sensors, protection elements and elements of holographic solar concentrators [76,77,78,79,80]

For holographic solar concentrator systems using photopolymers, the literature presents the use of commercial photopolymers (Bayfol HX200), acrylamide-based photopolymers, and acrylate-based photopolymers. These holographic solar concentrators are applied to concentrate solar spectrum radiation in PV and PVT systems. In addition, the wavelength for recording depends on the dye used in the composition of the photopolymer. For Bayfol HX200 has a combination of dyes that make it sensitive to virtually the entire visible spectrum, having its maximum sensitivity in the red range (633 nm). For acrylamide-based photopolymers, yellowish eosin (YE), which is sensitive to the green region (maximum sensitivity at 532 nm), is usually used. And finally, Biophotopol, which is an acrylate-based photopolymer, usually uses Riboflavin (RF) as a dye, which is sensitive to the blue range (over 450 nm). The following are the advances of these holographic concentrators and some relevant characteristics to be taken into account: the maximum modulation of the refractive index of the recording medium ($∆n_{max}$), hologram thickness (*T*), the wavelength of the recording beam (λ), spatial frequency (SF), diffraction efficiency (η), acceptance angle ($\theta_{acc}$), and the strategy to improve the acceptance angle.

### 4.1. HLs Using Photopolymer: Bayfol HX200

Bayfol HX200 photopolymer is in commercial use, and holographic elements can be developed that work in reflection and transmission mode; their characteristics are described by Bruder et al. [81]. Table 2 presents some relevant parameters to be taken into account.

In 2013, Altmeyer et al., reported their research results on the multiplexing of four diffractive gratings with a thickness of 16 μm each, behaving as a thick transmission hologram. This hologram configuration allowed them to redirect sunlight to a focal point with an acceptance angle of 20 degrees, and an average diffraction efficiency of 50%. The refractive index modulation was significant and ranged from 0.0067 to 0.032, making this photopolymer a very versatile recording medium. This research opened the door for photopolymers to be used as HLs in solar energy applications [82].

In 2016, Lee et al., proposed a multiplexed HL for three different incident angles using a 16 μm thick photopolymer. For recording, they used a convex lens with a focal length of 200 mm as the recording object and an incident beam intensity of 1.5 mW/cm^2^ at a wavelength of 532 nm. The reported results for the diffraction efficiency at each of the angles are approximately 71.14%, 73.95%, and 69.02% with a solar concentration efficiency of 26.73%, 35.31%, and 22.78%, respectively, as shown in Figure 9. These angles were chosen at three different times of the day from 10 am to 2 pm [83].

In 2017, Shaji Sam et al., developed the optimization of an HL for solar applications using an off-axis geometry, in registering the interference of a spherical wave acting as the object with a plane wave generated by the reference beam. For HL recording they used a 639 nm red laser with an intensity of 2 mW/cm^2^, which allowed them to optimize the HL, resulting in a diffraction efficiency of 91% with a photopolymer thickness of 16 μm [84].

In 2018, Marín-Sáez et al., proposed the use of a cylindrical HL to concentrate solar rays on a PV. For the HL recording stage, they used a 532 nm wavelength laser with varying intensities. They characterized the diffraction efficiency of the three cylindrical HLs constructed as 76%, 79%, and 72%, and obtained a maximum acceptance angle of 36.5°. As a suggestion, they proposed to include experimental photovoltaic tests with these HLs, as well as PV temperature studies [85].

In 2018, Wu et al., multiplexed three spherical lenses resulting in an HL that allowed them to redirect sunlight at a single fixed point. This system is capable of solar tracking for angles of 30°, 35°, and 40°; the diffraction efficiencies for each of these angles are approximately 72%, 74%, and 70%, respectively. The photopolymer thickness for the recording stage was 16.8 μm, with an average refractive index of the photopolymer of 1.58 [86].

In 2019, Marín-Sáez et al. designed and optimized an integrated system for solar protection of building facades. In this system, they used two non-multiplexed cylindrical HLs to focus solar radiation onto a PV, where they studied the effects of external environmental conditions on photopolymer aging and thermal effects on the PV. The system follows the apparent motion of the sun, with an acceptance angle of 72°. The recording stage for each of the cylindrical HLs was performed with a wavelength of 532 nm, and a photopolymer thickness of 16 μm with a maximum modulation of the refractive index of 0.0236 [87].

In 2020, Kao et al., proposed a stacked double-layer HL system that concentrates solar radiation onto two PVs. This concentrating system can reach an operating angle of 30°, and obtain a high average diffraction efficiency of 87.37%, with an operating wavelength bandwidth from 317 to 757 nm. For the registration of each of these HLs, they used a green laser with a wavelength of 532 nm and a photopolymer thickness of 16 μm, with a maximum refractive index modulation of 0.017 and a spatial frequency of 820 lines/mm [88].

### 4.2. HLs Using Acrylamide-Based Photopolymers

Within the photopolymerization process, one of the decisions to consider is the use of molecules that contribute to the initiation/sensitization of the holographic recording material for the formation of polymer chains. Depending on the molecule, this will produce changes in the reaction rate between the initiator and the monomer, due to the fact that the sensitivity of the photopolymer materials is related to these changes in the dye/initiator system [89]. Acrylamide is one of the most monomers used in photopolymer production, contributing to the modulation of the refractive index. Increasing the concentration of acrylamide improves the reaction rates in the photopolymerization process [90]. This polymeric material is stable over time and can be used in the production of HLs for concentrating solar power applications [91]. However, acrylamide is a toxic and carcinogenic compound [92]. Table 3 summarizes the most important parameters of holographic solar concentrators recorded on acrylamide-based photopolymers.

In 2013, Izabela Naydenova et al., presented an optical setup for recording diffraction gratings with different spatial frequencies, ranging from 450 to 1700 lines/mm, and with the photopolymer thickness of 50 μm, and multiplexed cylindrical lenses for application in solar energy concentrators with an average diffraction efficiency of about 52%. The wavelength for the recording stage was 633 nm with intensity modifications to find the optimum HL recording [93].

In 2014, Akbari et al., developed diffraction gratings to collect solar radiation at different times of the day, allowing to follow the apparent movements of the sun and concentrate the solar radiation in a PV, with an acceptance range of approximately 21°, this was possible to the combination of stacked gratings, which behave as a multiplexed HL performance and as a result for the diffraction efficiency they obtained 80%. The photopolymer thickness for recording was 75 μm, spatial frequency of 200 lines/mm, recording wavelength of 633 nm with intensity of 2 mW/cm^2^ [94]. For the same year, Akbari et al. presented another work, this time recording high-efficiency HLs, then stacking them in three layers on top of each other, making the incident sunlight concentrated at a point with an angular acceptance of 12° and a diffraction efficiency of 80% and 50%. For the recording stage, they used a thickness of 50 μm, wavelength of 633 nm, and spatial frequency of 300 lines/mm [95].

In 2015. Bianco et al., design and manufacture HLs for solar concentrator applications, the system consists of three multiplexed holographic lenses with an average diffraction efficiency of approximately 42% and an angular acceptance in each HL of 16°, in order to focus solar radiation on a PV. As for the HL recording stage, they used a photopolymer with a thickness of 30 μm, 0.02 refractive index modulation, and a 532 nm wavelength laser [96].

In 2016, Sreebha et al., etched a transmission HL for the purpose of using it in a glass window to serve as a sunlight concentrator on a PV, reporting a performance efficiency of 46%. For the recording stage, they performed an experimental setup off-axis and using a He-Ne laser with wavelength 633 nm with varying intensities [97].

In 2017, Akbari et al., performed the recording of two HLs, obtaining a maximum diffraction efficiency of 95% in each of these HLs and with an acceptance angle of 6° for each HL. These HLs allow concentrating the solar radiation on a PV, improving the performance of the PV system, which makes it a promising new technology. For the experimental setup, they performed it with off-axis geometry, using a laser wavelength of 532 nm and photopolymer thickness of 50 μm with a spatial frequency of 300 lines/mm, furthermore, they detailed the composition of the acrylamide-based photopolymer [98].

In 2018, Aswathy et al., presented the recording process and characterization of three HLs, demonstrating that they are suitable for solar energy applications allowing them to obtain significant yields in increasing PV electrical current intensities. For the multiplexing of the HLs, they used a variable time programming method to match the diffraction efficiency of the three lenses, obtaining an average diffraction efficiency of 19% and a maximum acceptance angle of 20°. For the recording stage, they used a laser with a wavelength of 633 nm, photopolymer thickness of 130 μm with a spatial frequency of 275 lines/mm [99].

In 2021, Ferrara et al., recorded and characterized different HLs in on- and off-axis configurations to obtain a set of five partially overlapping holographic lenses, achieving an acceptance angle of 80°. The HL system allowed concentrating the solar radiation on a PV continuously, without the need for mechanical movement to the relative positions of the sun. A laser with a wavelength of 532 nm, a photopolymer thickness of 125 μm, and a refractive index modulation of 0.02 was used for the recording stage [72].

### 4.3. HLs Using Acrylate-Based Photopolymers

Photopolymers based on acrylate monomers tend to use water-soluble constituents and are generally more difficult to prepare but more robust and environmentally friendly than acrylamide-based photopolymers. The low-toxicity photopolymer Biophotopol is another promising photopolymer for etching volumetric phase transmission HLs. This photopolymer was developed in 2007 by Ortuño et al. [100] and during the last few years has been optimized for different thicknesses. For the preparation of photopolymer, the following compounds are joined together: poly(vinyl alcohol) (PVA) as an inert binding polymer, sodium acrylate (NaAO) as a polymerizable monomer, triethanolamine (TEA) as a co-initiator and plasticiser, and sodium salt 5’-riboflavin monophosphate (RF) as a sensitizing dye [14,74,100,101]. Table 4 presents some relevant characteristics to be taken into account.

In 2018, Lloret et al. developed for the first time the recording of an HL in a photopolymer called Biophotopol which they developed in their research laboratory [14]. The recording and reconstruction setup was performed off-axis with a symmetric and asymmetric geometry, using a recording wavelength of 633 nm, photopolymer thickness of 300 μm, and spatial frequency of 1205 lines/mm.

In 2021, Morales-Vidal et al. presented the characterization of HLs etched in Biophotopol [102]. Showing the benefits of this photopolymer compared to other photopolymers, where they highlight: the ease of varying the physical thickness in the material, good diffraction efficiency, and wide acceptance angles in phase volume transmission holograms. They evaluated the performance of the HLs with an electronic assembly connected to a PV solar cell and a high-intensity solar simulator emitting a standard solar spectrum (AM1.5G).

In 2022, Morales-Vidal et al. presented multiplexed off-axis HL etching with symmetrical and asymmetrical low-frequency geometry (545 lp/mm) to concentrate sunlight from sunrise to sunset [103,104]. The efficiency of the complete “HLs-PV” system was evaluated for different angles of incidence see Figure 10.

They succeeded in multiplexing seven spherical lenses in Biophotopol material with a physical thickness of 197 μm; obtaining values in diffraction efficiency of approximately 50% and a wide acceptance angle of 60° (due to the angular separation of the multiplexed HL the current intensity is not constant in the angular range). In Figure 11 it can be seen a solar concentrating system of two holographic elements with seven HL multiplexed in each element.

Moreover, in 2024, Lloret et al. [73] optimized a holographic solar concentrator based on five multiplexed holographic lenses on Biophotopol. They reached a higher diffraction efficiency of 85% reducing the peak-to-peak distance up to 3.00°. The variation of the short-circuit current from peak to valley was negligible and the total acceptance angle achieved was 104°. The HSC design was formed by four elements.

## 5. Discussion

Among the three most commonly used recording materials for the production of HL in concentrating solar power, DCG and photopolymers stand out, according to published reviews [12,28,80,105,106,107,108,109,110]. Thin films with thicknesses of 2–3 μm and refractive index modulation values ∼0.080 can be obtained with DCG. The material and the hologram must be sealed, so that they can have an extended lifetime of approximately 25 to 30 years. However, as mentioned above, containing AD makes it an undesirable material, as it contains a carcinogenic, mutagenic, and reproductive toxic element [111]. On the other hand, photopolymers offer a wide range of thicknesses, ease of processing after the recording stage with good diffraction efficiency, and can be environmentally friendly.

Figure 12 shows a scheme with the diffraction efficiency and acceptance angle results of the different researchers so far, for the different photopolymers (Bayfol HX200, acrylamide-based photopolymers, and acrylate-based photopolymers). First, it can be seen that the fundamental difference between Bayfol HX200 and the other photopolymers is thickness. The available thickness is restricted, currently, available material thicknesses are approximately between 16 to 30 μm. Bayfol HX200 achieves higher acceptance angles when working with single elements, i.e., with 1 HL or with 2 HLs. Due to its easy handling, it also makes it a good candidate for the stacking technique. However, thicknesses would be appropriate for more multiplexing capability to be used in HL applications to concentrate solar energy [112]. The other two non-commercial acrylamide-based and acrylate-based photopolymers can be made for the necessary thickness conditions in applications that require them, these conditions make them suitable as recording materials for multiplexed HLs in concentrating solar energy. However, studies report that the use of acrylamide is probably carcinogenic in humans and has neurological and reproductive effects [113,114], so the big difference between these two is that the acrylate based ones are more environmentally friendly and less toxic. In both cases, they can be optimized for different dyes and different thicknesses, i.e., they are easier to modify chemically. It can be seen that the trend is towards acrylate-based photopolymers based on the multiplexing technique.

Figure 13 shows a scheme of the HSCs time evolution. They are classified according to the strategy used to optimize the important parameters, diffraction efficiency (red circle), and acceptance angle (blue circle). It can be seen that when the HSC is composed of a single HL, the diffraction efficiency is very high, but the acceptance angle is low. With overlapping and stacking techniques one can try to optimize these parameters, but it is also difficult. If two HLs are used, it can be observed that the acceptance angle increases, and the diffraction efficiency is also relatively high. Finally, it is observed that to achieve high DE and large AA, the multiplexing technique is the most effective. At first, HSCs with high DE but small AA were achieved. Recently, however, HSCs with high DE and wide AA have been achieved.

Finally, Figure 14 shows a comparison in terms of efficacy, durability, cost-effectiveness, and nontoxicity properties for Bayfol HX200 (blue), acrylamide-based photopolymers (red), and acrylate-based photopolymers (green). The established comparison criteria have been defined from 0 (not at all) to 10 (very much). It can be seen that, in terms of efficiency and durability, the three photopolymers reach the highest value on the scale. However, the comparison in terms of cost-effectiveness shows a large difference between Bayfol HX200 (commercial photopolymer) and the more economically attractive photopolymers based on PVA/Acrylamide and PVA/Acrylate. Finally, the comparison related to non-toxicity is also significant. A large difference can be seen among the three polymers.

Although acrylamide-based photopolymer is a good holographic photopolymer likely it will be abandoned in the long term due to its high toxicity. At the same time, and due to its low toxicity, the use of acrylate-based photopolymers will increase in the next years. The properties and characteristics that scientists should pay attention to in acrylate-based photopolymers are those that will lead to better results in the intended application. The trend in solar concentration will probably be towards the use of crosslinkers and copolymers to optimize the properties of acrylate polymers and achieve better results in terms of wide acceptance angle and high diffraction efficiency. For all these reasons, the promising photopolymer for future research in the development of HLs for concentrated solar power is acrylate-based, as it has low toxicity [115,116,117,118], such as Biophotopol.

## 6. Conclusions

The development of new technologies must be in harmony with the environment; therefore, one of the global objectives is to make the energy transition towards clean energies, such as wind, solar, tidal, etc. In this review, the recording materials for developing diffraction gratings or HL that are applied to concentrate solar radiation in a PV or PVT system, especially photopolymers, were detailed, highlighting the characteristics necessary for the development of these technologies.

The main objective of developing HLs is to concentrate solar energy into a system that transforms that energy into another type of energy, for example, electricity. These HLs must overcome the limits of conventional solar concentrators, such as tracking solar radiation during the day, overheating of the PVs to receive the full solar spectrum, the high costs of these, and the space occupied by the system. For HLs to be able to track solar radiation throughout the day, it is necessary to multiplex several holograms in the same material, so that it acts for different ranges of inclinations and thus be able to diffract the solar spectrum during all day in the desired location. However, so far, the literature does not yet record a single multiplexed HL capable of tracking the entire solar spectrum during the day. To overcome this drawback, several cascaded HLs have been proposed to obtain a complete tracking of the solar spectrum radiation. For this reason, it is important to choose an appropriate material for the development of HLs that meets the following conditions: low toxicity, ease of fabrication, low cost, and that meets the characteristics of an HL (good angular selectivity, spectral selectivity, stability over time, and high diffraction efficiency). According to the above, acrylate-based photopolymers are one of the most compliant with these conditions, making it attractive for future research, in terms of the ability to multiplex and optimize the largest number of spherical and cylindrical lenses possible to record, as well as to study its stability over time to ensure its useful life.

Finally, multiplexed HLs allow for chromatic selectivity, which is essential for future hybrid system designs, allowing the use of all solar radiation and thus increasing the efficiency of the system, e.g., to generate electricity and thermal energy simultaneously. Therefore, the current challenges, limitations and prospects of photopolymers used in HSC is the optimization of the photopolymer composition in order to increase the dynamic range to be able to multiplex more HLs in the same hologram while maintaining high diffraction efficiency and thus increase the acceptance angle even more.

## Figures and Tables

**Figure 1 polymers-16-00732-f001:**
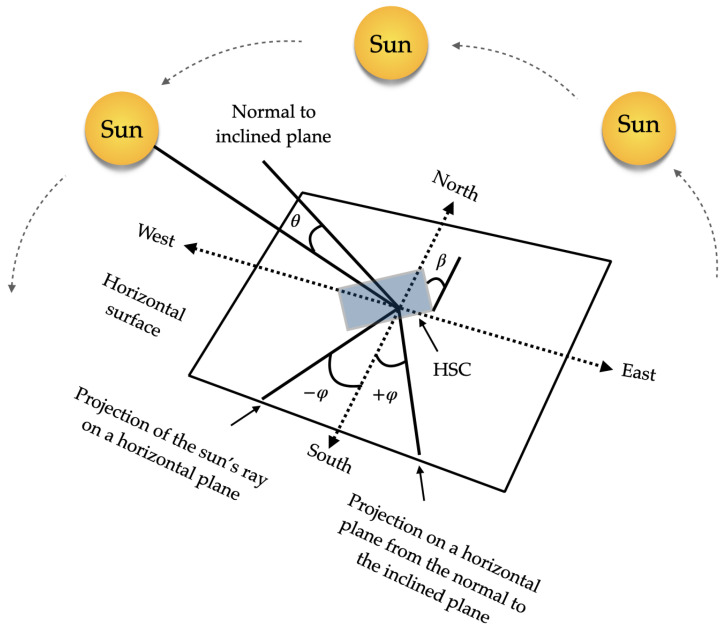
Representation of solar radiation with respect to the receiving surface.

**Figure 2 polymers-16-00732-f002:**
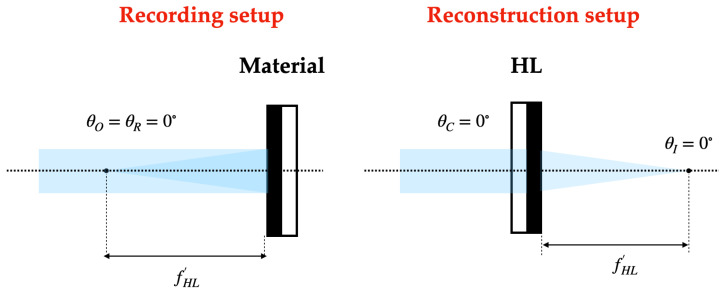
On-axis holographic setup.

**Figure 3 polymers-16-00732-f003:**
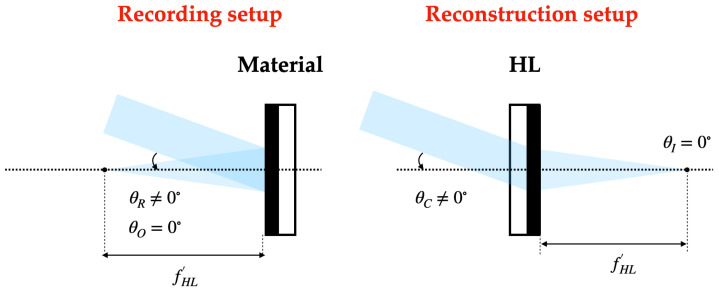
Asymmetrical off-axis holographic setup.

**Figure 4 polymers-16-00732-f004:**
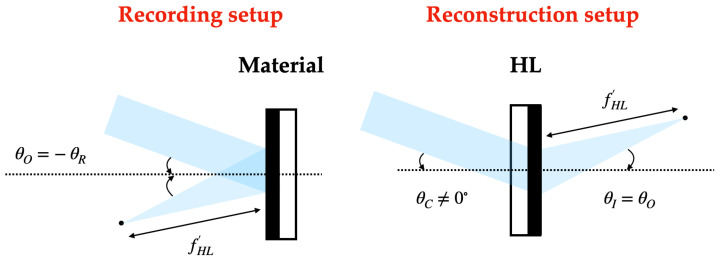
Symmetrical off-axis holographic setup.

**Figure 5 polymers-16-00732-f005:**
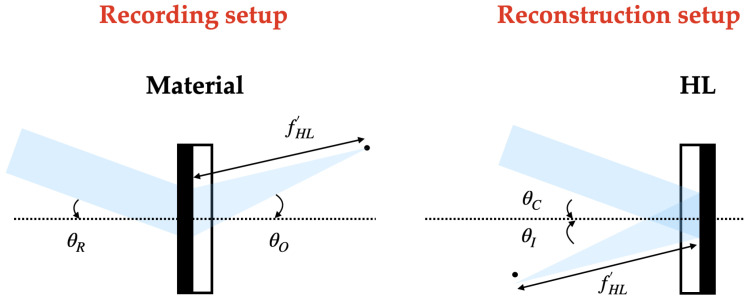
Holographic setup for the recording and reconstruction of reflection HLs.

**Figure 6 polymers-16-00732-f006:**
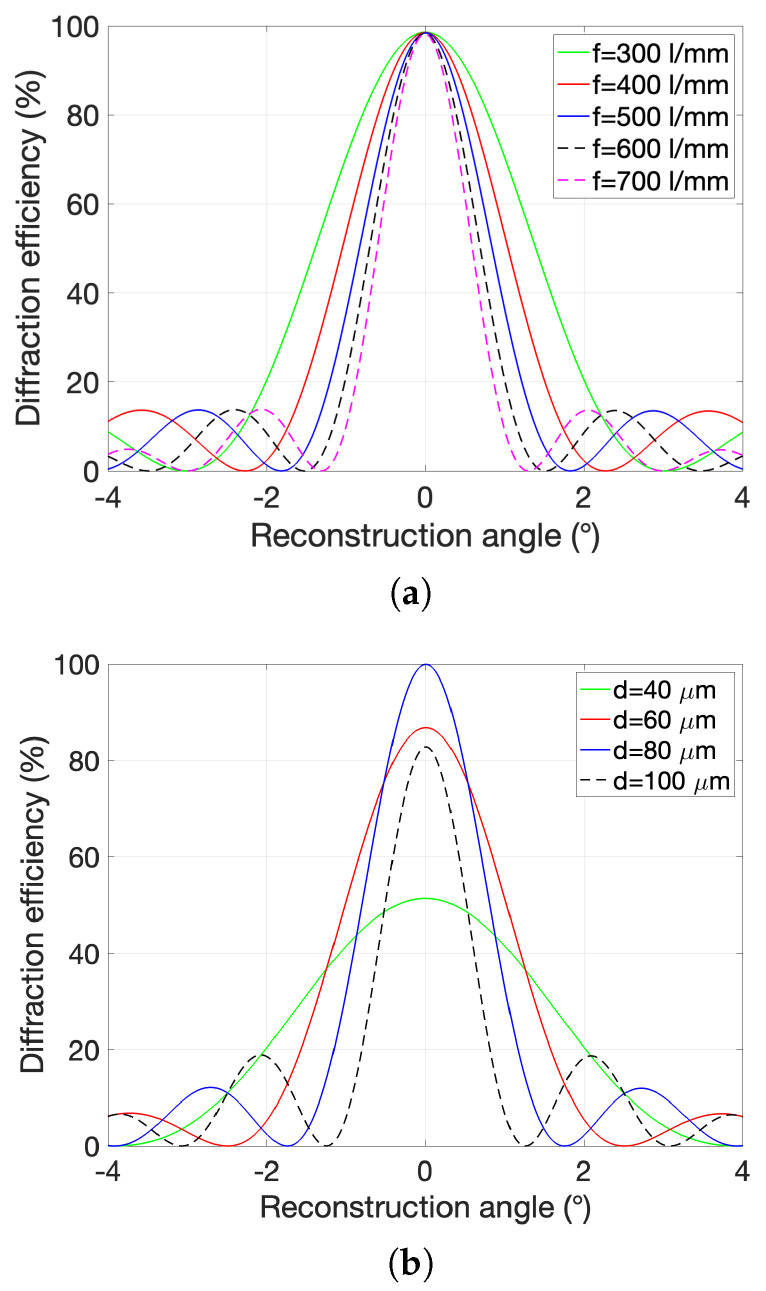
Theoretical results based on Kogelnik’s theory for transmission holograms. (**a**) Theoretical comparison between volume holograms with different frequencies. Considered $n_{1}=0.004$ and d=60
μm. (**b**) Theoretical comparison between volume holograms with different optical thicknesses (*d*). Considered $n_{1}=0.004$, f=500 l/mm.

**Figure 7 polymers-16-00732-f007:**
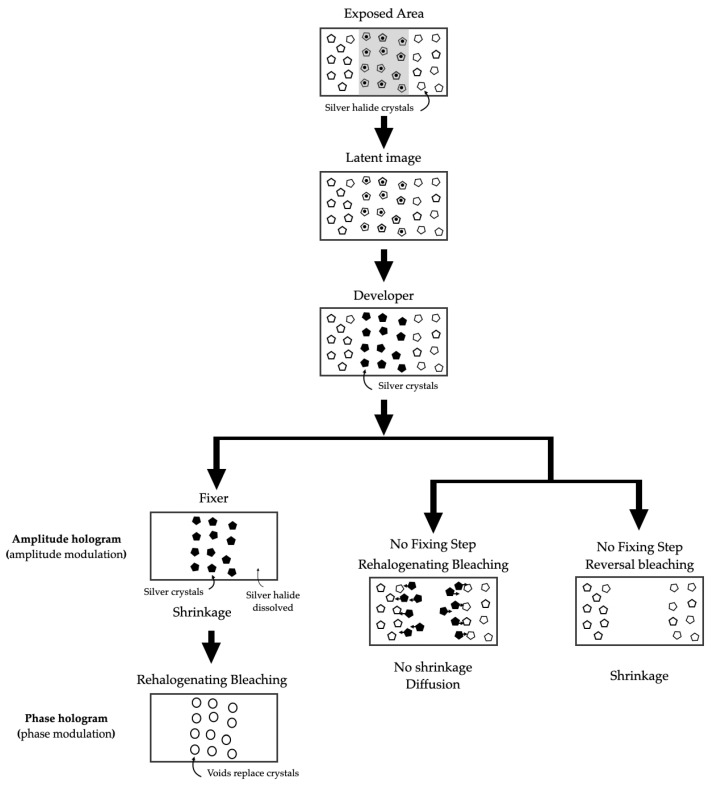
Chemical process for hologram recording in silver halide materials.

**Figure 8 polymers-16-00732-f008:**
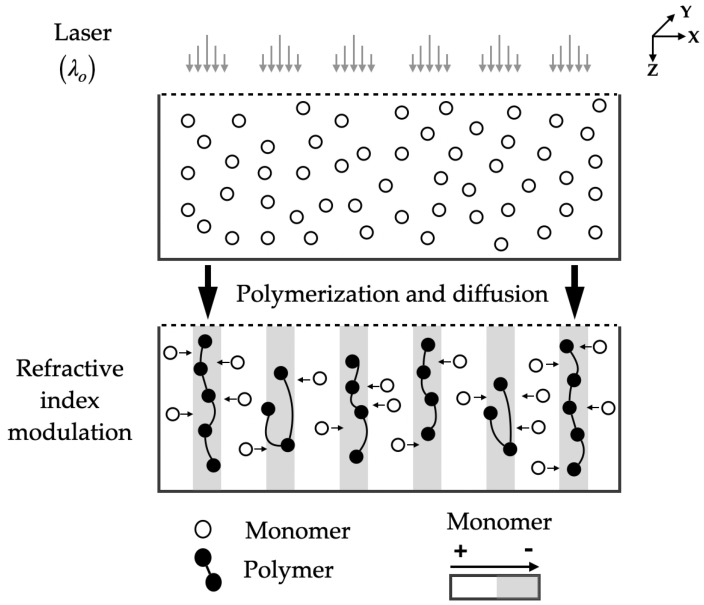
Interferential pattern storage of a phase hologram by refractive index modulation in a photopolymer.

**Figure 9 polymers-16-00732-f009:**
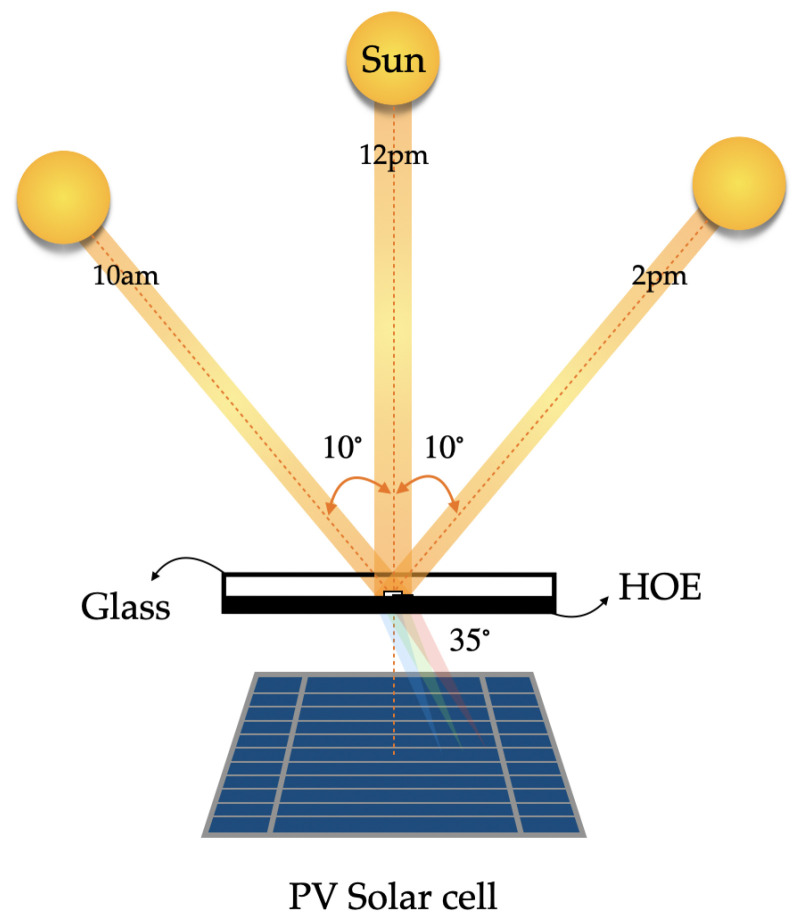
Schematic diagram of a holographic solar concentrator with off-axis angular multiplexing.

**Figure 10 polymers-16-00732-f010:**
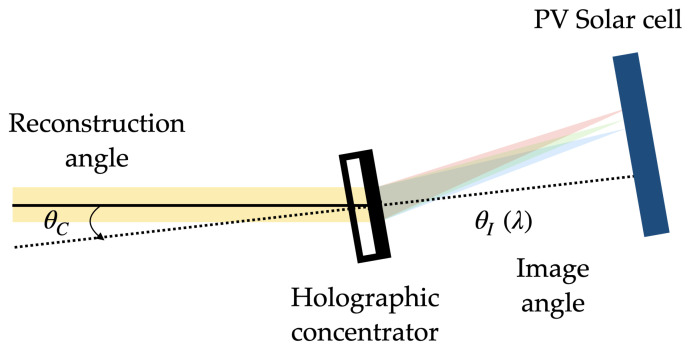
Representation scheme for the solar concentrating system of a HL and PV.

**Figure 11 polymers-16-00732-f011:**
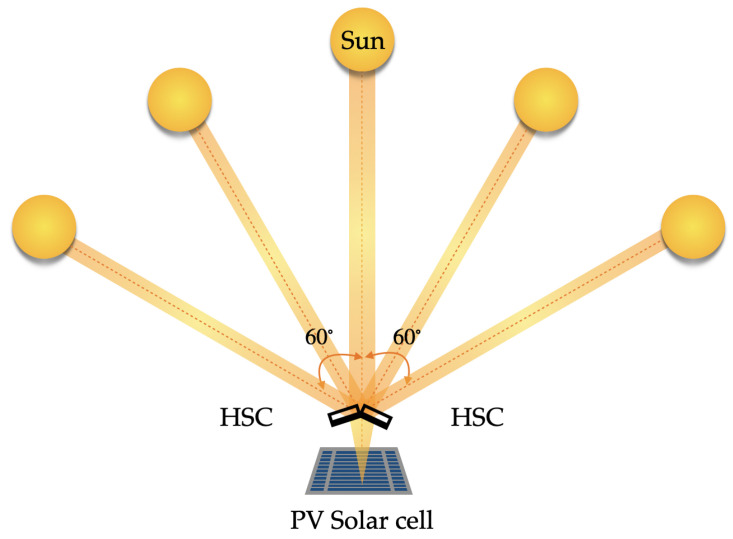
Representation scheme for the solar concentrating system of a HL and PV.

**Figure 12 polymers-16-00732-f012:**
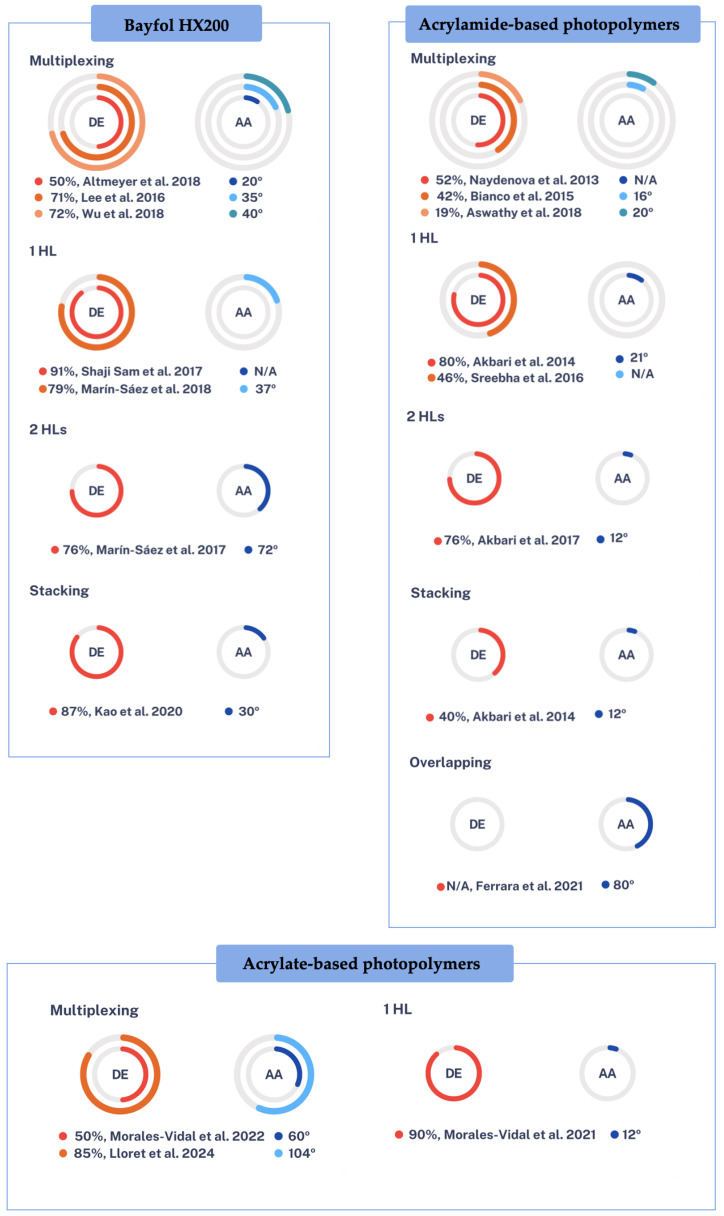
Outline of the results obtained so far for HSCs as a function of the photopolymers and strategies used [72,73,82,83,84,85,86,87,88,93,94,95,96,97,98,99,102,103].

**Figure 13 polymers-16-00732-f013:**
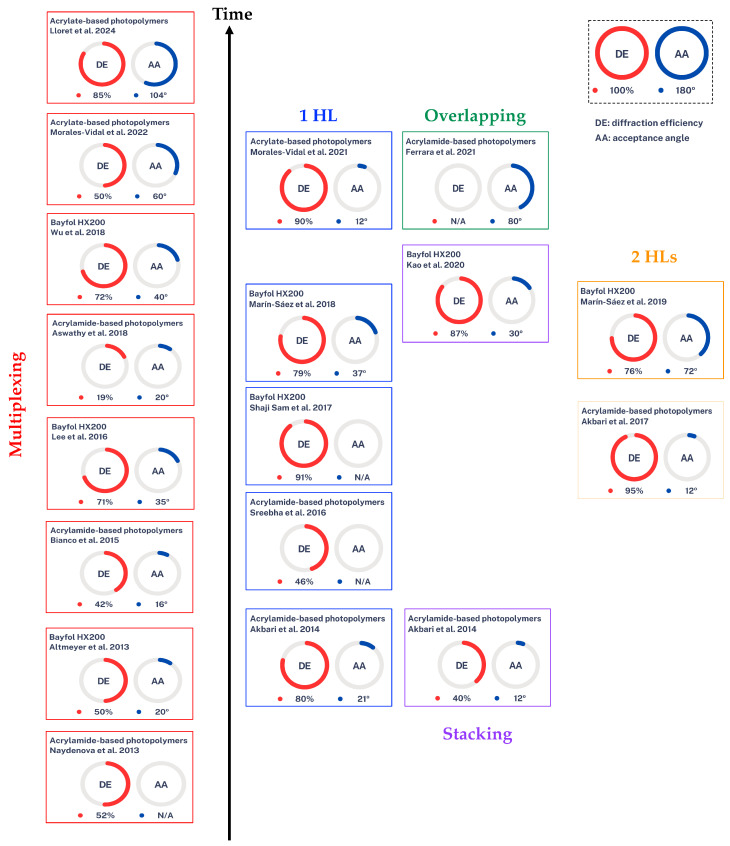
Outline of the different strategies used to optimize the most important parameters in the design of holographic solar concentrators. DE: diffraction efficiency, AA: acceptance angle [72,73,82,83,84,85,86,87,88,93,94,95,96,97,98,99,102,103].

**Figure 14 polymers-16-00732-f014:**
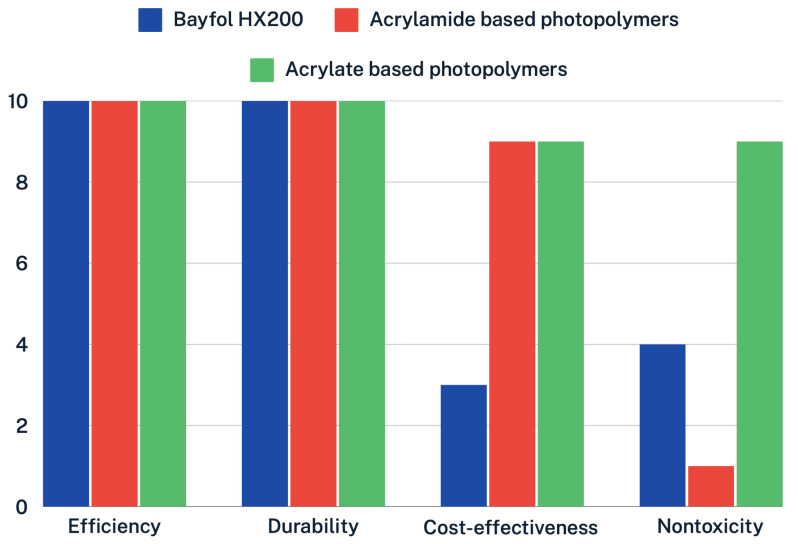
Comparison in terms of efficiency, durability, cost-effectiveness, and nontoxicity for Bayfol HX200 (blue), acrylamide-based photopolymers (red), and acrylate-based photopolymers (green). The comparison criteria are defined from 0 (not at all) to 10 (very much).

**Table 1 polymers-16-00732-t001:** Details of some characteristics of the most commonly used holographic recording materials in solar energy applications.

Recording Material	$\eta_{\max}$ (%)	$\theta_{\mathrm{acc}}$ (°)	Stability	Multiplexing Capability	Spectral Sensitivity (mJ/cm^2^)	Spectral Response (nm)
Silver halide [63,64,65,66,67]	50 to 90	35	N/A	Yes	$10^{-5}$ to $10^{-3}$	250 to 700
DCG [65,66,68,69,70,71]	60 to 95	100	Yes	Yes	10 to $10^{3}$	350 to 580
Photopolymers [65,66,72,73]	46 to 95	104	Yes	Yes	1 to $10^{3}$	300 to 1100

**Table 2 polymers-16-00732-t002:** Holographic solar concentrators based on Bayfol HX200 and their relevant characteristics.

Author	$∆n_{\max}$	*T* (μm)	λ (nm)	SF (lines/mm)	η (%)	$\theta_{\mathrm{acc}}$ (°)	Strategy
Altmeyer [82]	0.0067 to 0.032	16	532	N/A	50	20	Multiplexing
Lee [83]	N/A	16	532	N/A	71	35	Multiplexing
Shaji Sam [84]	0.03	16	639	1080	91	N/A	1 HL
Marín-Sáez [85]	N/A	16	532	70 to 1200	79	37	1 HL
Wu [86]	N/A	16	532	N/A	72	40	Multiplexing
Marín-Sáez [87]	0.0190 to 0.0236	16	532	N/A	76	72	2 HLs
Kao [88]	0.017	16	532	820	87	30	Stacking

**Table 3 polymers-16-00732-t003:** Holographic solar concentrators using acrylamide-based photopolymer and their relevant characteristics.

Author	$∆n_{\max}$	*T* (μm)	λ (nm)	SF (lines/mm)	η(%)	$\theta_{\mathrm{acc}}$ (°)	Strategy
Naydenova [93]	N/A	50	633	450–1700	52 (average)	N/A	Multiplexing
Akbari [94]	N/A	75	633	200	80	21	1 HL
Akbari [95]	N/A	50	633	300	80:50	12	Stacking
Bianco [96]	0.02	30	532	N/A	42 (average)	16	Multiplexing
Sreebha [97]	N/A	N/A	633	N/A	46	N/A	1 HL
Akbari [98]	N/A	50	532	300	95	12	2 HLs
Aswathy [99]	N/A	130	633	275	19 (average)	20	Multiplexing
Ferrara [72]	0.02	125	532	N/A	N/A	80	Overlapping

**Table 4 polymers-16-00732-t004:** Holographic solar concentrators using acrylate-based photopolymer and their relevant characteristics.

Author	$∆n_{\max}$	*T* (μm)	λ (nm)	SF (lines/mm)	η(%)	$\theta_{\mathrm{acc}}$ (°)	Strategy
Lloret [14]	N/A	300	633	1205	N/A	N/A	1 HL
Morales-Vidal [102]	N/A	90	633	922	90	12	1 HL
Morales-Vidal [103]	0.00425	197	633	545	50 (average)	60	Multiplexing
Lloret [73]	0.004 to 0.0042	185	633	525	85 (average)	104 *	Multiplexing

* This angle was achieved from an HSC based on 4 holograms, each consisting of 5 multiplexed holographic lenses with an acceptance angle of 26°.

## Data Availability

The supporting information can be found from the corresponding author.

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
