# Peer review of "Photopolymer Holographic Lenses for Solar Energy Applications: A Review"

_polymers, 2024, doi:10.3390/polym16060732_

Round 1

Reviewer 1 Report

Comments and Suggestions for Authors

1] The acrylate-based photopolymers are highlighted as one of the most suitable materials for HL development, Authors should compare with other existing materials in terms of efficiency, durability, and cost-effectiveness for readers better understanding.

2] Authors should elaborate with suitable references on how thickness influences the multiplexing capability and overall performance of holographic lenses in concentrating solar power applications?

3] Authors should highlight the current challenges, limitations and prospects associated with the use of photopolymers in holographic solar concentrator systems.

Reviewer 2 Report

Comments and Suggestions for Authors

Authors -  Eder Alfaro, Tomás Lloret, Juan M. Vilardy, Marlón Bastidas, Marta Morales-Vidal, Inmaculada Pascual

 Manuscript « Photopolymer holographic lenses for solar energy applications: a

review» - they are experts in the field of optical recording. Some clarifications are also needed.

The review is of great fundamental importance, since it concerns not only the chemistry of polymers, but also the physical foundations of recording and creating holographic lenses.

1.     Since the journal is mainly about polymers and their properties, I would like to see a conclusion about which structures should be sought to work with. The authors mention polymers based on acrylamide, PVA. What is the reason for this choice? What characteristics do these polymers have, that only they are mentioned in the article. And Bayfol HX photopolymer.

2.     In what wavelength range is it preferable to record?

3.     if acrylamide-based polymers are toxic, and the message of the review is precisely to reduce environmental harm, then can we conclude that this may cause the abandonment of these polymers? And if the answer is yes, what properties and characteristics of new polymer materials should scientists pay attention to? I think this is a very important point that should be reflected in the conclusions.
